# Isoniazid Preventive Therapy for Prevention of Tuberculosis among People Living with HIV in Ethiopia: A Systematic Review of Implementation and Impacts

**DOI:** 10.3390/ijerph20010621

**Published:** 2022-12-29

**Authors:** Dawit Getachew Assefa, Eden Dagnachew Zeleke, Delayehu Bekele, Dawit A. Ejigu, Wondwosen Molla, Tigist Tekle Woldesenbet, Amdehiwot Aynalem, Mesfin Abebe, Andualem Mebratu, Tsegahun Manyazewal

**Affiliations:** 1Center for Innovative Drug Development and Therapeutic Trials for Africa (CDT-Africa), College of Health Sciences, Addis Ababa University, Addis Ababa P.O. Box 3880, Ethiopia; 2Department of Nursing, College of Medicine and Health Sciences, Dilla University, Dilla P.O. Box 419, Ethiopia; 3Department of Midwifery, College of Health Science, Bule-Hora University, Bule-Hora P.O. Box 144, Ethiopia; 4Department of Obstetrics and Gynecology, Saint Paul’s Hospital Millennium Medical College, Addis Ababa P.O. Box 3880, Ethiopia; 5Department of Pharmacology, Saint Paul’s Hospital Millennium Medical College, Addis Ababa P.O. Box 3880, Ethiopia; 6Department of Midwifery, College of Medicine and Health Sciences, Dilla University, Dilla P.O. Box 419, Ethiopia; 7Department of Public Health, School of Graduate Studies, Pharma College, Hawassa P.O. Box 5, Ethiopia; 8School of Nursing, College of Medicine and Health Sciences, Hawassa University, Hawassa P.O. Box 1560, Ethiopia

**Keywords:** isoniazid prevention therapy (IPT), HIV, tuberculosis, antiretroviral therapy (ART), Ethiopia

## Abstract

Background: Tuberculosis (TB) is a major cause of morbidity and mortality in people living with HIV (PLWHIV). Isoniazid preventive therapy (IPT) prevents TB in PLWHIV, but estimates of its effects and actual implementation vary across countries. We reviewed studies that examined the impact of IPT on PLHIV and the factors influencing its implementation in Ethiopia. Methods: We searched PubMed/MEDLINE, Embase, and the Cochrane Central Register of Clinical Controlled Trials from their inception to 1 April 2021 for studies of any design that examined the impact of IPT on PLHIV and the factors influencing its implementation. The protocol was registered in PROSPERO, ID: CRD42021256579. Result: Of the initial 546 studies identified, 13 of which enrolled 12,426 participants, 15,640 PLHIV and 62 HIV clinical care providers were included. PLHIV who were on IPT, independently or simultaneously with ART, were less likely to develop TB than those without IPT. IPT interventions had a significant association with improved CD4 count and reduced all-cause mortality. IPT was less effective in people with advanced HIV infection. The major factors influencing IPT implementation and uptake were stock-outs, fear of developing isoniazid-resistant TB, patient’s refusal and non-adherence, and improper counseling and low commitment of HIV clinical care providers. Conclusion: IPT alone or in combination with ART significantly reduces the incidence of TB and mortality in PLHIV in Ethiopia than those without IPT. More research on safety is needed, especially on women with HIV who receive a combination of IPT and ART. Additionally, studies need to be conducted to investigate the efficacy and safety of the new TPT (3 months combination of isoniazid and rifapentine) in children and people living with HIV.

## 1. Introduction

Tuberculosis (TB) remains one of the top 10 causes of death globally and the primary cause of death from a single infectious agent [1]. In 2021, there were 10.6 million TB cases globally and 1.4 million deaths among HIV-negative people, and an additional 187,000 deaths among HIV-positive people [1]. Most people who acquired TB in 2021 were in the regions of South-East Asia (45%), Africa (23%), and the Western Pacific (18%) [1]. In Africa, a widespread scale-up of antiretroviral therapy (ART) strongly declines the incidence of TB [2]. Ethiopia is one of the top 14 triple burden countries for TB, TB/HIV, and MDR-TB [1]. An estimated incidence of all forms of TB in Ethiopia, in 2019, was 140/100,000 population, with 111,039 TB cases notified [1]. TB remains one of the major causes of morbidity and mortality in the country [3].

To prevent and reduce the incidence of TB in people living with HIV (PLHIV), the World Health Organization (WHO) recommended the use of isoniazid preventive therapy (IPT) as a mainstay of the “Three I’s” approach [4]. Per the WHO recommendation, IPT is administered at a daily dose of a maximum of 300 mg daily for 6–9 months in adults and adolescents and 5 mg/kg for children [5,6]. This chemoprophylaxis reduces the risk of an early episode of TB occurrence in people with latent infection or those exposed to infection, and reduces recurrent episodes of TB [5,6]. For patients with latent TB, IPT can be beneficial, with the potential to reduce TB infection irrespective of HIV status [5,6] and protecting communities from [7,8,9]. However, the emergence of drug resistance secondary to IPT administration is a potential risk that is understudied [5].

In Ethiopia, the National TB guideline recommends that IPT should be provided to all HIV-infected individuals who are unlikely to have active TB irrespective of CD4 count, ART status, pregnancy status, or history of treatment for a prior episode of TB before three years. Patients should be supported at home level either by local health extension workers or a family supporter to ensure daily administration of IPT. Patients should be given a one-month supply of isoniazid for six months, with a monthly scheduled follow-up integrated with other treatment services. IPT should also be administered for asymptomatic children under five who were exposed to TB within the past year. However, the current evidence regarding the effects and actual implementation of IPT in Ethiopia remains unclear. This systematic review was, therefore, conducted to examine the outcomes of IPT in PLHIV and factors influencing its implementation in Ethiopia.

## 2. Methods

The protocol for this systematic review and meta-analysis was registered at the International Prospective Register of Systematic Reviews (PROSPERO) database, ID: CRD42021256579. The Preferred Reporting Items for Systematic Review and Meta-Analysis (PRISMA 2020) guidelines were followed to choose studies to be included in this review.

### 2.1. Eligibility Critria Studies

Criteria included studies of any design that examined the impact of IPT on PLHIV and the factors influencing its implementation in Ethiopia, published in English language until 1 April 2021.

### 2.2. Participants

Patients with HIV,Man or woman of any age,ART naïve or experienced on their time of enrollment.

### 2.3. Types of Interventions

#### 2.3.1. For Intervention Studies

Intervention group: isoniazid 300 mg daily for 6–9 months in adults and adolescents and 5 mg/kg for children.Comparison group: inactive placebo, ART only, or no preventive treatment.

#### 2.3.2. For Non-Intervention Studies

Effectiveness, barriers, or opportunities implementing IPT program in Ethiopia based on experiences from PLHIV or HIV clinical service providers.

### 2.4. Outcome Measures

#### 2.4.1. Primary Outcome Measure

Incidence of definite or probable TB. Definite TB was defined by a microbiological, chest X-ray, or histological identification of TB.

#### 2.4.2. Secondary Outcome Measures

Incidence of death.Factors associated with TB incidence.Incidence of adverse drug reactions leading treatment discontinuation.Barriers to implementation of IPT.

### 2.5. Search Strategy

A computerized systematic search method was used to search for articles from online databases PubMed/MEDLINE, Embase, and Cochrane Central Register of Clinical controlled Trials (CENTRAL) databases from inception to 1 April 2021. The search was based on the *Cochrane Handbook for Systematic Reviews of Interventions* [10], with a combination of the words (tuberculosis) AND (isoniazid preventive therapy) considered the major search term (Table 1).

### 2.6. Study Selection

The *Cochrane Handbook for Systematic Reviews of Interventions* [11] was followed. To import the research articles from the electronic databases and remove duplicates, ENDNOTE software version X7 was used. Two authors independently reviewed the results of the literature search and obtained full-text copies of all potentially relevant studies. Disagreements were resolved through discussion. When clarification was necessary, the trial authors were contacted for further information. The screening and selection process was reported in a PRISMA flow chart (Figure 1).

### 2.7. Data Extraction and Management

The title and abstract were produced from the electronic search and were independently screened by two authors based on studies and the selection criteria. The information collected were study characteristics, including study design, study setting, age and number of participants enrolled, interventions, study title, journal, year of publication, publication status, follow-up period, funding of the trial or sources of support, baseline characteristics of study subjects, incidence of TB, factors associated with IPT, and IPT implementation barriers. One author independently extracted data, and these were cross-checked by another author. Missing data were requested from the authors whenever necessary.

## 3. Result

A total of 546 studies through the databases were searched, of which, 16 full-text studies were assessed further for eligibility and 13 of them fulfilled the inclusion criteria for further analysis (Table 2).

### 3.1. Characteristics of Included Studies

In this review, 13 studies that enrolled a total of 15,640 PLHIV and 62 health professionals working on HIV care were included (Table 2)**.** Many of the patients were aged greater than 14 years [12,13,14,15,16] and female [12,13,14,15,17,18]. The CD4 count was > 200 cell count/μL in most of the participants in one study [13] and less than 199 cells/mm^3^ in another study [14], and most of the patients were in HIV stage III followed by stage II [13,14,19].

### 3.2. Incidence of TB

Among PLHIV, the incidence of TB after administration of IPT independently or simultaneously with ART shows a significant reduction in the incidence of TB as compared to patients who were on ‘ART only’ and ‘No intervention’ [12,13,15,16,17,18,19,20,21]. Simultaneous administration of both IPT and ART reduced the incidence of TB by 80% [12], 93.7% [13], and 65% [14], respectively. Completion of IPT showed a significant protective effect against the occurrence of active TB for 3 years when compared to IPT non-exposed patients [13]. IPT was associated with a significant change in CD4 count [15,19,20] and reduced all-cause mortality [14,17,19,20].

### 3.3. Factors Associated with TB Incidence among PLWHIV Who Took IPT

The risk of developing TB or dying was significantly higher in PLHIV on WHO stage III and above at baseline [12,13,14,16,17,18,22], male [12,13,14,22], with a CD4 count of less than a 350 cell count/μL and those with opportunistic infections [15,16,17,18,21,22], children with delayed motor development [21] who did not take cotrimoxazole preventive therapy [18,21,22], use anti-pain [22], and have a hemoglobin level less than 10 mg/dL [16,21,22]. The risk of TB infection and death was lower in those who held good body weight [14,18,21,22] and referred to the hospital from other health facilities [14]. In some of the studies, the effects of age [14,21] and baseline CD4 count had a suboptimal effect on TB incidence or death [14].

### 3.4. Barriers in the Implementation of IPT

A significant number of HIV clinical care providers reported that several barriers hinder IPT coverage and its effective implementation, including isoniazid stock-outs, fear of developing isoniazid resistance, patient’s refusal and non-adherence, and improper counseling and low commitment of HIV clinical care providers [23]. Lack of patient empowerment and proper counseling on IPT, weak patient/healthcare provider communication, information gaps, low commitments from health administrators and other stakeholders to effectively run the IPT program, and underlying mental health issues resulting in missed or irregular patient adherence to IPT were also reported as barriers for effective implementation of IPT in Ethiopia [24]. Additionally, clinician impressions that ruling out active TB among HIV patients is difficult was found to be a significant barrier to IPT uptake [25].

**Table 2 ijerph-20-00621-t002:** Characteristics of included studies.

S. No	Study ID	Design	Setting	Age	Follow Up	Subjects		Patient Important Outcome	
1.	Mindachew et al., 2014 [24]	Qualitative	Hospital	N/A	N/A	12 heath professional		barriers	Lack of patient empowerment and proper counseling on IPT, weak patient/healthcare provider relationship, lack of patient information, low reinforcement by health officials and stakeholders to strengthen IPT uptake and adherence forgetfulness, patient IPT non-adherence, and non-disclosure of HIV zero-status.
2.	Yirdaw et al., 2014 [12]	Retrospective cohort	Hospitals (n = 5)	Mean (30 years)	2 years	5407 patients		IPT before ART	aHR = 0.18, 95% CI = 0.08–0.42
								IPT before ART’	HR = 0.25, 95% CI = 0.11–0.59
								IPT and ART	aHR = 0.20,95% CI = 0.10–0.42
								IPT and ART	HR = 0.36; 95% CI = 0.17–0.74
								IPT only	HR = 0.24, 95% CI = 0.13–0.44
								IPT after ART	HR = 0.19, 95% CI = 0.11–0.34
								TB incidence	295
3.	Assebe et al., 2015 [17]	Retrospective cohort study	Hospital	N/A	Mean 24.1 months	588		Overall TB incidence	49
						IPT 294	No IPT 294	Overall TB incidence	3.78 cases per 100 PY (95% CI: 2.85, 4.99 cases per 100 PY)
								Incidence of TB among IPT Plus ART	2.22 cases per 100 PY (95% CI: 1.29, 3.82 cases per 100 PY)
								Incidence of TB among ART alone	5.06 cases per 100 PY (95% CI: 3.65, 7.02 cases per100 PY)
								Incidence of TB among IPT Plus ART	aHR 2.02 (95% CI: 1.04–3.92)
4.	Nigusse et al., 2015 [16]	Retrospective follow up study	Hospital	Median 38 (IQR: 31.2–42)	5 years	480		Overall TB incidence	70
								Overall TB incidence	3.59 per 100 PY
								TB incidence among IPT	aHR = 0.49, 95% CI = 0.26–0.94
5.	Ayele, 2015 [14]	Retrospective cohort study	Hospital	Range 15–99 years	839 days	1922 (374 received IPT)		Overall TB incidence	258
								Incidence of TB/death among IPT plus ART	HR = 0.35; 95% CI (0.16, 0.77)
								Incidence of TB/death among ART alone	HR = 1.22; 95% CI (0.45, 3.28)
								Incidence of TB/death among IPT plus ART	aHR = 0.40; 95% CI (0.18, 0.87)
								Incidence of TB among IPT plus ART	5.20 per 100 PYs
								Incidence of TB among ART alone	8.05 per 100 PYs
6.	Alemu et al., 2016 [21]	Retrospective cohort study	Hospitals (n = 2), Health centers (n = 6)	Median (IQR) 6 (3.5–9.00) years	N/A	645		Overall TB incidence	79
								Overall TB incidence	4.2: 95% CI (3.4, 5.3) PY
7.	Teklay et al., 2016 [23]	Qualitative study	Hospitals (n = 11)	Mean (±SD) 30 (±6) years	N/A	50 health providers		Barriers	Isoniazid stock out
									Fear of creating isoniazid resistance Problems in patient acceptance Lack of commitment of health managers
8.	Abossie et al., 2017 [15]	Hospital-based retrospective study	Hospital	Mean (±SD) 31.27 (+12.0)		271		Incidence of TB among IPT Plus ART	12 (8.7%)
						IPT 138	No IPT 133	Incidence of TB among ART alone	37 (27.8%)
								Incidence of TB among IPT Plus ART	RR 0.31 (95% CI 0.122, 0.49)
9.	Semu et al., 2017 [13]	Retrospective cohort	Public health institutions	Mean (±SD) 34.9 (±9.1) years	5 years	2524 patients		TB Incidence Rate among IPT	0.21/100 PY
								TB-incidence Rate among at IPT completion	aIRR 0.037 (95% CI, 0.016–0.072)
								overall TB incidence	6.7/100 PY
								TB incidence	277
								Incidence of TB among IPT-with-HAART	0.42/100 PY
								Incidence of TB among IPT-with-HAART	aIRR = 0.063 (95% CI 0.035–0.104)
								Incidence of TB among alone HAART	7.83 cases/100 PY
10.	Tiruneh et al., 2019 [18]	Retrospective cohort study	Hospital and health center	Mean (±SD) 33 years (±9) years	Median 26 months	600			
						IPT 200	No IPT 400	Overall TB Incidence	53 (8.8%)
								Overall TB Incidence	57 cases per 100 PY
								Incidence of TB among IPT group	1.98 per 100 PY
								Incidence of TB among non-IPT group	4.52 per 100 PY
								Incidence of TB among IPT group	aHR 0.45, 95% CI 0.219–0.920
								Incidence of TB among IPT group	HR 0.397, 95% CI 0.203–0.774
11.	Gebremariam et al., 2020 [20]	Retrospective cohort study	Hospitals (n = 2)	N/A	5 years	968 patients		Incidence of TB among ART plus IPT	8 (0.5 cases/100 PY)
						IPT 484	No IPT 484	Incidence of TB among ART plus IPT	aHR 0.17; 95% CI 0.08–0.35
								Incidence of TB among ART alone	49 (3 cases/100 PY)
								Deaths on ART plus IPT	12 (0.5 cases/100 PYs)
								Deaths on ART alone	35 (2.1 cases/100 PYs)
								Death reduction among ART plus IPT	aHR 0.48; 95% CI 0.24–0.97
12.	Atey et al., 2020 [19]	Retrospective Cohort Study	Hospitals (n = 5)	N/A	N/A	1863		Incidence of TB among IPT Plus ART	28
						IPT 621	No IPT 1242	Incidence of TB among ART alone	272
								Overall incidence	300
								Incidence rate of mortality among IPT Plus ART	440 per 100,000 PYs
								Incidence rate of mortality among ART alone	1490 per 100,000 PYs
13.	Legese et al., 2020 [22]	Institutional based cross-sectional	Hospital	Mean (±SD) 37.94 (±12.15)	6 months	372 (231 on IPT)		Overall incidence of TB among IPT group	13 (3.5%)

## 4. Discussion

In this study, IPT shows a significant reduction in the risk of TB and dying as compared to that of ART only and non-intervention. It also shows an effect on the improvement of CD4 count. Several studies conducted elsewhere also reported that IPT reduces the incidence of TB [26,27,28,29,30]. Recent studies reported that IPT is effective in the reduction in TB disease on pregnant women living with HIV and with their CD4 count ≤ 350 cells/μL [31,32]. In West Africa, the early initiation of ART and 6 months of IPT showed a significant reduction in HIV-related illness by 44% and the risk of mortality from any cause by 35% as compared to the risks with deferred initiation of ART and no IPT [33]. There were several factors, such as being male, low baseline CD4 count, and hemoglobin level less than 10 mg/dl, that negatively influenced the effectiveness of IPT. In agreement with our finding, a study from Malawi reported that male PLHIV had low adherence to IPT as compared to female [34]. This might be a reason for the difference in the effectiveness of IPT between male and female patients. Additionally, the protective effect of INH was more extreme in contacts exposed to drug-sensitive tuberculosis (adjusted hazard ratio, 0.30; 95% confidence interval, 0.18–0.48) and to multidrug-resistant tuberculosis (adjusted hazard ratio, 0.19; 95% confidence interval, 0.05–0.66) compared with those exposed to mono-INH-resistant tuberculosis (adjusted hazard ratio, 0.80; 95% confidence interval, 0.23–2.80) [35].

Implementation of the IPT program in Ethiopia is facing several challenges, including stock-out that may question sustainability of the program and may provoke drug resistance. In concurrence to our finding, studies from India reported the lack of awareness on the role and way of taking IPT, risk perception among patients’ parents, cumbersome screening process, isoniazid stock-outs, inadequate knowledge among healthcare providers, and poor programmatic monitoring as main barriers to IPT implementation [36,37]. Adequate supply and availability of isoniazid at the health facilities, preparing unambiguous treatment guideline, contact tracing, provision of IPT for children, community-based intervention, and provision of adequate training for health care providers on IPT enhanced the reduction in TB incidence and patient’s adherence to IPT [38,39,40,41]. Thus, for effective implementation and outcomes of IPT program in Ethiopia, there is a need to enhance patient adherence and effectiveness of IPT through effective communication, build the capacity of the healthcare providers through training and motivation packages, sustainably increase isoniazid supply, and strengthen program partnership and collaboration.

## 5. Conclusions

IPT alone or in combination with ART significantly reduces the incidence of TB and mortality in PLHIV in Ethiopia than those without IPT. More research on safety is needed, especially on women with HIV who receive a combination of IPT and ART. Additionally, studies need to be conducted to investigate the efficacy and safety of the new TPT (3-month combination of isoniazid and rifapentine) in children and people living with HIV.

## Figures and Tables

**Figure 1 ijerph-20-00621-f001:**
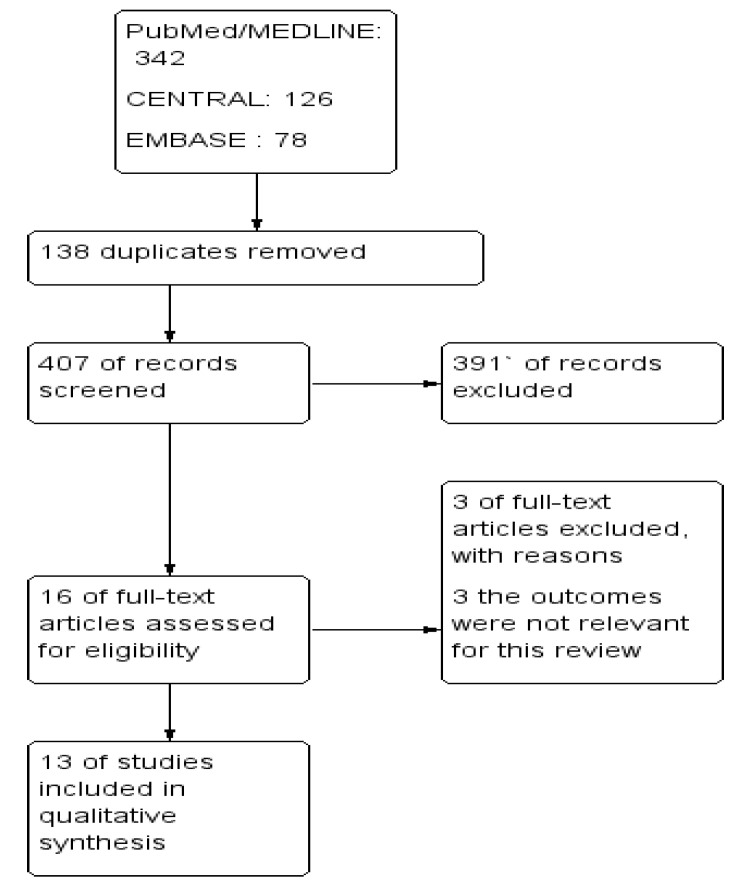
PRISMA study flow diagram.

**Table 1 ijerph-20-00621-t001:** Search term for MEDLINE.

Search	Most Reset Queries
#1	Search HIV infections[MeSH] OR HIV[TW] OR HIV-1*[MeSH] OR HIV-2*[TW] OR HIV-1[TW] OR HIV-2[TW] OR HIV infec* [TW] OR Human immunodeficiency virus [TW] OR Human immune-deficiency virus [TW] OR ((Human immune* [TW]) AND (deficiency virus [TW])) OR Acquired immune deficiency syndrome [TW] OR AIDS [MeSH] OR ((acquired immune* [TW]) AND (deficiency syndrome [TW])) OR sexual transmitted diseases, viral [MeSH: No Exp]
#2	Search Tuberculosis [MeSH] OR TB [MeSH]
#3	Search preventive therapy [MeSH] OR Chemoprevention [MeSH] OR Prophylaxis [MeSH]
#4	Search #1 AND #2 AND #3
#5	Search #1 AND #2 AND #3 Limits: Publication date from 1980 to 2021

## Data Availability

All relevant data are within the manuscript.

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
