# Peer review of "Isoniazid Preventive Therapy for Prevention of Tuberculosis among People Living with HIV in Ethiopia: A Systematic Review of Implementation and Impacts"

_ijerph, 2022, doi:10.3390/ijerph20010621_

Round 1

Reviewer 1 Report

I think the use of meta-analysis is proper in this article. However, there were certain problem within the literature search and interpretation. Also, there are some places can be improved to make it more readable. I suggest the authors consider the following modification and clarifications.

1. Introduction. WHO report had the latest TB data (2021 WHO report). It should be refreshed. Also, No. 55-5668-70 sentences were not complete.

2.The involved 13 studies haven’t been clearly referred to and marked in the context. I tried my best to identify them with their original publication.

(1) Studies 1,4,7,8,10,11 and 13 seem appropriate with their interpretation.

(2) Study 2, the protective effect of “IPT before ART” was “aIRR 0.25(0.11-0.59)” in the table2, but it was “HR 0.25(0.11-0.59) in the original article. For study 3,5,6, it was “per 100 PYs” instead “per 100 PY”, “er 100 PYs” or ”PYs”. For study 9, incidence of TB among alone HAART was “7.83/100PYs” instead “7.83 ases/100PYs” in the table2.

(3) Studies 2 and 12, the sample size and results were wrong in table 2 (not consistent with original studies).

(4) Study 5, the outcomes of hazard were TB or death in original studynot just the hazard

of TB incidence in table 2.

3.As a meta-analysis for the barriers of IPT implementation, it might not be enough to involve only two qualitative articles. There were many reasons that impede implementation, but this article only included experiences from people living with HIV or HIV clinical providers. I do suggest authors may consider more relative studies, e.g. Lai J, Dememew Z, Jerene D, et al. Provider barriers to the uptake of isoniazid preventive therapy among people living with HIV in Ethiopia. Int J Tuberc Lung Dis. 2019;23(3):371-377. doi:10.5588/ijtld.18.0378

Author Response

Response to Reviewer 1 Comments

Point 1: I think the use of meta-analysis is proper in this article. However, there were certain problem within the literature search and interpretation. Also, there are some places can be improved to make it more readable. I suggest the authors consider the following modification and clarifications.

Response 1: Thank you for insightfull comment. We first planned to include meta-analysis, but as you saw on the tabel, the authors report there result in so many different ways(HR,RR and so on). That is why we didn’t do meta-analysis.

Point 2: Introduction. WHO report had the latest TB data (2021 WHO report). It should be refreshed. Also, No. 55-56、68-70 sentences were not complete.

Response 2: Thank you for insightfull comment. We now corrected accordingly.

Point 3: The involved 13 studies haven’t been clearly referred to and marked in the context. I tried my best to identify them with their original publication.

(1) Studies 1,4,7,8,10,11 and 13 seem appropriate with their interpretation.

(2) Study 2, the protective effect of “IPT before ART” was “aIRR 0.25(0.11-0.59)” in the table2, but it was “HR 0.25(0.11-0.59) in the original article.

Response 3: Thank you for insightfull comment. We now corrected accordingly.

For study 3,5,6, it was “per 100 PYs” instead “per 100 PY”, “er 100 PYs” or ”PYs”.

Response 4: Thank you for insightfull comment. We now corrected accordingly. But, some of the studies reported as per 100 PYs.

For study 9, incidence of TB among alone HAART was “7.83/100PYs” instead “7.83 ases/100PYs” in the table2.

Response 5: Thank you for insightfull comment. We now corrected accordingly.

(3) Studies 2 and 12, the sample size and results were wrong in table 2 (not consistent with original studies).

Response 6: Thank you for insightfull comment. We now corrected accordingly. But, the sample size for study 12 was correctly described.

(4) Study 5, the outcomes of hazard were TB or death in original study not just the hazard

of TB incidence in table 2)

Response 7: Thank you for insightfull comment. We now corrected accordingly.

As a meta-analysis for the barriers of IPT implementation, it might not be enough to involve only two qualitative articles. There were many reasons that impede implementation, but this article only included experiences from people living with HIV or HIV clinical providers. I do suggest authors may consider more relative studies, e.g. Lai J, Dememew Z, Jerene D, et al. Provider barriers to the uptake of isoniazid preventive therapy among people living with HIV in Ethiopia. Int J Tuberc Lung Dis. 2019;23(3):371-377. doi:10.5588/ijtld.18.0378

Response 8: Thank you for insightfull comment. We now corrected accordingly.

Reviewer 2 Report

1. Concerning the design of the study, I would suggest to add a  minimum number of participants in a in the criteria of selected publications. There should be a statistical significance of the sample size.  Is it possible to substantiate the statistical significance of the result obtained on 12 participants?

2. How much do you have Isoniazid-resistance TB cases in Ephiopia? How effective might be INH preventive therapy for originally INH-resistant M. tuberculosis strains? I think that this question should be discussed.

3. The text of manuscript contains typos such as merged words or word wrap sign in the middle of the line (for instance, line 25 exam-ined, etc.), this should be fixed. 

4. Page 3 line 126: what does "Ionized" therapy mean? Is it a mistake?

5. Page 4, lines 138 and 152 - a references should be added

Author Response

Response to Reviewer 1 Comments

Point 1: Concerning the design of the study, I would suggest to add a  minimum number of participants in a in the criteria of selected publications. There should be a statistical significance of the sample size.  Is it possible to substantiate the statistical significance of the result obtained on 12 participants?

Response 1: Thank you for your comments. Since our study is systematic review, 12 is the number of studies not the participants. These 12 studies we included in our review have enrolled 15,640 study participants.

Point 2: How much do you have Isoniazid-resistance TB cases in Ethiopia? How effective might be INH preventive therapy for originally INH-resistant M. tuberculosis strains? I think that this question should be discussed.

Response 2: Thank you for your comments. We couldn’t able to find a specific study conducted in Ethiopia, but we have discussed with other similar latest study which was conducted in another country.

Point 3: The text of manuscript contains typos such as merged words or word wrap sign in the middle of the line (for instance, line 25 exam-ined, etc.), this should be fixed. 

Response 3: Thank you for your comments. We now corrected accordingly.

Point 4:  Page 3 line 126: what does "Ionized" therapy mean? Is it a mistake?

Response 4: Thank you for your comments. We now corrected accordingly.

Point 5:  Page 4, lines 138 and 152 - a references should be added

Response 4: Thank you for your comments. We now corrected accordingly.

Reviewer 3 Report

The paper reviews papers that evaluate implementation and impact of isoniazid preventive therapy (IPT) for prevention of tuberculosis among people living with HIV (PLHIV) in Ethiopia. The authors find that IPT alone or in combination with ART significantly reduces the incidence of TB and mortality in PLHIV, but that more research is needed.

The research question is relevant and the methods are generally relevant. The search strategy and the searched databases seem appropriate. Also, the main finding of a reduction of TB incidence among IPT treated persons relative to non-IPT-treated persons is justified.

However, my main concerns with this paper are 1. that presentation of results is not very clear, and 2. that presentation of results of the two other questions (factors associated with TB incidence and Barriers in the implementation of IPT) is not necessarily justified from results.

Information in the main table, Table 2, is not very systematic nor clear. E.g. age of participants is given both by mean and median and in some instances by IQR or range. Time of data collection is not given, which would be helpful. Outcomes are given in many formats that are not easy to understand, E.g. Overall TB incidence (S. no. 3) is given both as ‘49’ and as 3.78 cases per 100 PYs. Which is which? In S. No. 8 Incidence is given as 12 (8.7%). Is that 8.7% prevalence? It is difficult to compare findings from the different studies from the table and I suggest to make some kind of summarizing table or graph that directly show the same outcomes (e.g. TB in PLHIV +/- IPT) by the different studies in a directly (or more clear) comparative way.

I don’t think the results in sections 3.3 and 3.4 are quite justified on the table, as, with other words, the information in the table regarding these issues are quite sporadic and not very systematic. It is unclear whether the factors associated with TB incidence (section 3.3) is for Tb as such or whether they specifically negatively influence the effectiveness of IPT. This is not indicated in Section 3.3. but only in the Discussion.

Generally, findings regarding factors associated with TB incidence (and reduced effect of IPT) and barirers to the implementation are only described in qualitative terms. If figures for magnitude of reduced effects etc. exist it would be warranted to somehow show this which would improve the discussion of the effects of these factors.

Thus, I think the findings need to be presented more stringent and in a more comparable way, e.g. by better table(s). Also, section 3.3. (‘Factors associated with TB incidence’) need to be put more in context of IPT rather than just as risk factors for TB development.

Specific comments:

P. 2 L. 83: Aim: Unclear whether this concerns Ethiopia alone or other countries also. Suggest to state that this only concerns studies that deal with outcomes of IPT in Ethiopia. Also to make it clear that this is not an evaluation of IPT but of IPT in Ethiopia.

P. 4 L. 150: Result: 546 studies were searched, but only 16 were assessed further for eligibility, and of these 13 fulfilled the inclusion criteria. What were the main reasons for excluding the 546 – 16 = 530 studies?

P. 6: Table: Please see above to make this table more stringent with summaries of studies being comparable. Also include observation time.

P. 9: Sections 3.3 and 3.4: See above. Please describe these results in better accordance with Table 2.

P. 9: Discussion: Does not seem quite focused but list rather than discuss a number of factors related to IPT. A bit difficult to read. Suggest to make the discussion more focused based on findings.

P. 10: Conclusion: Hard to see that the statement ‘There needs arealistically attainable system in place…’ is supported from the findings.

Generally, there are many linguistic or type-set errors (e.g. P. 2, L 55: ‘In Africa, a widespread scale-up ofantiretroviral therapy (ART) strongly declines the incidence of [2]’. Ofantiretroviral is in one word and HIV (?) is missing). I suggest that the text is revised by a native English speaking person. Also, errors like P. 3 L 107: Berries? Barriers?

Author Response

Response to Reviewer 3 Comments

The paper reviews papers that evaluate implementation and impact of isoniazid preventive therapy (IPT) for prevention of tuberculosis among people living with HIV (PLHIV) in Ethiopia. The authors find that IPT alone or in combination with ART significantly reduces the incidence of TB and mortality in PLHIV, but that more research is needed.

The research question is relevant and the methods are generally relevant. The search strategy and the searched databases seem appropriate. Also, the main finding of a reduction of TB incidence among IPT treated persons relative to non-IPT-treated persons is justified.

Point 1: However, my main concerns with this paper are 1. that presentation of results is not very clear, and 2. that presentation of results of the two other questions (factors associated with TB incidence and Barriers in the implementation of IPT) is not necessarily justified from results.

Response 1: Thank you for your insightful comments. Since our design was systematic review. We systematically synthesized the report from the included studies using qualitative method. The included studies have reported their result using adjusted hazard ratio. We have taken those specific results and interpreted narratively. For the barrier in the implementation of IPT section we have summarized the results from two qualitative and one cross sectional studies.

Point 2: Information in the main table, Table 2, is not very systematic nor clear. E.g. age of participants is given both by mean and median and in some instances by IQR or range. Time of data collection is not given, which would be helpful. Outcomes are given in many formats that are not easy to understand, E.g. Overall TB incidence (S. no. 3) is given both as ‘49’ and as 3.78 cases per 100 PYs. Which is which? In S. No. 8 Incidence is given as 12 (8.7%). Is that 8.7% prevalence? It is difficult to compare findings from the different studies from the table and I suggest to make some kind of summarizing table or graph that directly show the same outcomes (e.g. TB in PLHIV +/- IPT) by the different studies in a directly (or more clear) comparative way.

Response 2: Thank you for your insightful comments. In the study characteristics table, we included what the included studies missioned.  Some studies described in mean and SD and some of them reported in median and IQR. Regarding study no-3, 49 is the overall incidence and 3.78 was the number of cases per 100 PYs. Furthermore, 12 is the number of cases out of the total population. We strongly believe that this table is the best way to describe the report.

Point 3: I don’t think the results in sections 3.3 and 3.4 are quite justified on the table, as, with other words, the information in the table regarding these issues are quite sporadic and not very systematic. It is unclear whether the factors associated with TB incidence (section 3.3) is for Tb as such or whether they specifically negatively influence the effectiveness of IPT. This is not indicated in Section 3.3. but only in the Discussion.

Response 3: Thank you for your insightful comments. Since it is the characteristics of included studies, we don’t think missioning all the results on the table necessary. That is why we only described on the result part. Our study aimed to assess the incidence of TB among HIV positive patients who either took IPT plus ART or ART alone.  The included studies reported the factors associated with the incidence of TB among patients who have taken IPT. Or those patients took IPT but after some time they have developed TB and several factors were associated with the incidence.

Point 4: Generally, findings regarding factors associated with TB incidence (and reduced effect of IPT) and barirers to the implementation are only described in qualitative terms. If figures for magnitude of reduced effects etc. exist it would be warranted to somehow show this which would improve the discussion of the effects of these factors.

Response 4: Thank you for your insightful comments. Those are the only figures we were able to get from our literature search.

Point 5: Thus, I think the findings need to be presented more stringent and in a more comparable way, e.g. by better table(s). Also, section 3.3. (‘Factors associated with TB incidence’) need to be put more in context of IPT rather than just as risk factors for TB development.

Response 5: Thank you for your insightful comments. Those associated factors were describing the incidence of TB among HIV positive patients who have taken IPT before. We now corrected accordingly.

Specific comments:

Point 6: P. 2 L. 83: Aim: Unclear whether this concerns Ethiopia alone or other countries also. Suggest to state that this only concerns studies that deal with outcomes of IPT in Ethiopia. Also to make it clear that this is not an evaluation of IPT but of IPT in Ethiopia.

Response 6: Thank you for your insightful comments. It is a concern for other countries, but since our study tries to assess the situation in Ethiopia. We want to evaluate IPT in Ethiopia. We now corrected accordingly.

Point 7: P. 4 L. 150: Result: 546 studies were searched, but only 16 were assessed further for eligibility, and of these 13 fulfilled the inclusion criteria. What were the main reasons for excluding the 546 – 16 = 530 studies?

Response 7: Thank you for your insightful comments. As we described on Figure one  138 were duplicates and the other studies were excluded because either the title or their outcome of interest did not match with our inclusion criteria.

Point 8: P. 6: Table: Please see above to make this table more stringent with summaries of studies being comparable. Also include observation time.

Response 8: Thank you for your insightful comments. We have described the follow-up time.

Point 9: P. 9: Sections 3.3 and 3.4: See above. Please describe these results in better accordance with Table 2.

Response 9: Thank you for your insightful comments. Since it is the characteristics of included studies, we don’t think missioning all the results on the table necessary. That is why we only described on the result part.

Point  10: P. 9: Discussion: Does not seem quite focused but list rather than discuss a number of factors related to IPT. A bit difficult to read. Suggest to make the discussion more focused based on findings.

Response 10: Thank you for your insightful comments. We now corrected accordingly.

Point 11: P. 10: Conclusion: Hard to see that the statement ‘There needs arealistically attainable system in place…’ is supported from the findings.

Response 11: Thank you for your insightful comments. We now corrected accordingly.

Point  12: Generally, there are many linguistic or type-set errors (e.g. P. 2, L 55: ‘In Africa, a widespread scale-up ofantiretroviral therapy (ART) strongly declines the incidence of [2]’. Ofantiretroviral is in one word and HIV (?) is missing). I suggest that the text is revised by a native English speaking person. Also, errors like P. 3 L 107: Berries? Barriers?

Response 12: Thank you for your insightful comments. We now corrected accordingly.

Round 2

Reviewer 1 Report

Table 2: Study 5. The outcome of 'Incidence of TB among ART alone' was still not corrected as ' 8.05 per 100 PYs'.

Study 12. The description in the original paper was 'Eligible patients on ART (n = 1, 863) were categorized into one-to-two ratios of exposed groups to IPT (n = 621) and nonexposed groups to IPT (n = 1, 242). ' So the cited sample size was reversed.

Author Response

Thank you. We now corrected accordingly. 
